# Charge carrier mobility in thin films of organic semiconductors by the gated van der Pauw method

Cedric Rolin[1], Enpu Kang[1], Jeong-Hwan Lee[1], Gustaaf Borghs[2], Paul Heremans[1,3] & Jan Genoe[1,3]

Thin film transistors based on high-mobility organic semiconductors are prone to contact problems that complicate the interpretation of their electrical characteristics and the extraction of important material parameters such as the charge carrier mobility. Here we report on the gated van der Pauw method for the simple and accurate determination of the electrical characteristics of thin semiconducting films, independently from contact effects. We test our method on thin films of seven high-mobility organic semiconductors of both polarities: device fabrication is fully compatible with common transistor process flows and device measurements deliver consistent and precise values for the charge carrier mobility and threshold voltage in the high-charge carrier density regime that is representative of transistor operation. The gated van der Pauw method is broadly applicable to thin films of semiconductors and enables a simple and clean parameter extraction independent from contact effects.

[1] IMEC, Large Area Electronics, Kapeldreef 75, Leuven B-3001, Belgium. [2] KU Leuven, Department of Physics and Astronomy, Celestijnenlaan 200d, Leuven B-3001, Belgium. [3] KU Leuven, Department of Electrical Engineering, Kasteelpark Arenberg 10, Leuven B-3001, Belgium. Correspondence and requests for materials should be addressed to C.R. (email: cedric.rolin@imec.be).

The charge carrier mobility is a key performance criteria for organic semiconductors[1]. High-mobility values allow fast device operation as needed for low-cost electronics on large areas with performance meeting market demands[2–5]. Mobility is conveniently extracted from thin film transistors (TFT) characteristics using the standard gradual channel approximation model[6,7]. This approach evaluates the mobility of charges during their transport through the high-density accumulation layer at the semiconductor-dielectric interface[8,9]. This value is therefore directly representative of transistor operation and is a relevant parameter for device integration into circuits[10,11].

In high-mobility organic semiconductors and in short channel devices, however, the relative importance of the contact resistance $R_c$ can be such that the standard model is no longer appropriate for mobility extraction[12,13]. Proper parameter extraction is complicated by the fact that carrier injection from the contact into the semiconductor is often mediated by the gate voltage $V_G$. When this is not properly taken into account, it leads to serious over-estimation of the mobility[14–16]. Therefore, a more accurate, yet simple, method is highly desirable for the proper evaluation of $\mu_{tfsc}$, the charge carrier mobility in thin films of organic semiconductors in the high-charge density accumulation layer. In this definition, $\mu_{tfsc}$ characterizes the contact-independent translational motion of charge carriers across the thin film semiconductor material, over distances that may be larger than typical grain size. In this sense, $\mu_{tfsc}$ encompasses extrinsic barriers to transport such as grain boundaries and therefore does not necessarily correspond to the intrinsic intra-grain charge carrier mobility of the monocrystalline semiconductor[17].

In this work, we proposed the gated van der Pauw (gVDP) method for the characterization of thin films of several organic semiconductors. The van der Pauw (VDP) method is a geometry-independent four-contact electrical measurement widely used to evaluate the sheet conductance $\sigma_s$ of thin continuous slabs of semiconductor materials[18,19]. The use of a gate to modulate charge density in VDP devices is, on the other hand, hardly documented in the literature[20–22]. In this communication, we show that the gate in the gVDP structure creates transport conditions similar to TFT operation. We propose a simple model for the interpretation of gVDP characteristics, allowing for an extraction of mobility and threshold voltage $V_T$. We then fabricate devices based on thin films of seven different organic semiconductors and show that their measurements are independent of $R_c$ and are representative of the electrical

characteristics of the thin film in the high-charge density regime. This validates the gVDP method as a simple and accurate technique to extract $\mu_{tfsc}$.

## Results

**The gated van der Pauw method.** A simple VDP device topology is presented in Fig. 1a. The thin conductive film is patterned as a square and four contacts are applied to its corners. Although the film shape can be arbitrary, semiconductor films with four-fold symmetry simplify data analysis and increase accuracy. In the structure of Fig. 1a, the size of the contacts must be negligible relative to the size of the square, but this condition is relaxed when using clover-leaf shaped films, which simplifies alignment of the patterned layers. For electrical measurement, a current $I_1$ is sourced in contact 1 and drained at contact 2, which is grounded and used as a reference. The potentials $V_3$ and $V_4$ in isolated contacts 3 and 4 are measured. The potential distribution and current density streamlines in a square VDP device are obtained from a two-dimensional finite element analysis solving Maxwell's equations with realistic $\sigma_s = 1.5\,\mu S$ per square and $I_1 = 1\,\mu A$ (Fig. 1c). The current density is highest along the edge 1–2 and decreases towards side 3–4 as the current path lengthens. The dashed equipotential lines $V_3$ and $V_4$ delimit the probed region where the voltage is sensed, away from the source and drain contacts 1 and 2.

For $\sigma_s$ extraction, we first measure $R_{12} = \frac{|V_3 - V_4|}{I_1}$, the resistance in the probed region alongside 1–2. The measurement of $R_{21}$ along the same side is obtained by reversing the direction of the current while grounding contact 1. Next, the resistances along the three other sides of the square are measured in a similar way and the eight resistance values are averaged as $\bar{R}$. Finally, thanks to the four-fold symmetry of the film, the sheet conductance of the film is simply obtained as $\sigma_s = \frac{\ln(2)}{\pi \bar{R}}$. Strong points of the VDP method are the following. As in four point probe (FPP) measurements, the contacts that sense voltage are non-injecting, thereby limiting contact effects. Furthermore, contrarily to the FPP method, no geometrical dimension enters VDP data analysis: device imperfections and misalignments are averaged out by measuring all four sides in both directions.

A common gate is introduced by fabricating the VDP device on a highly doped silicon wafer covered with a thin layer of $SiO_2$ as gate dielectric. A gVDP device cross-section is shown in Fig. 1b, where all contacts are projected on the same plane for

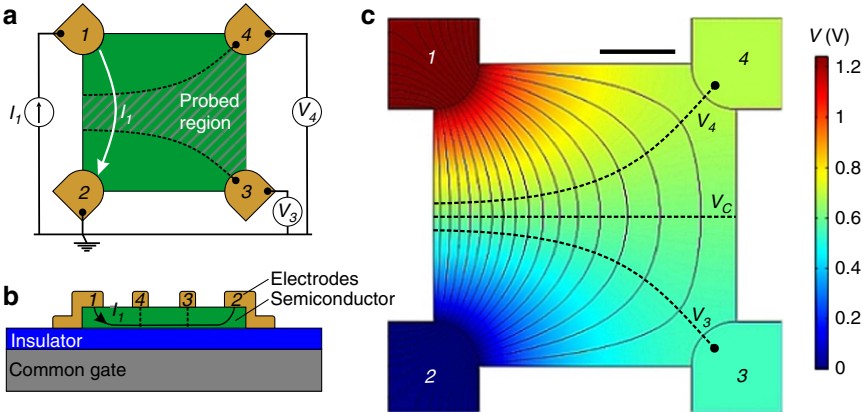

**Figure 1 | The gated van der Pauw method.** (**a**) Top view of a van der Pauw device with square-shaped thin semiconductor film. (**b**) Pseudo-cross-section view of a van der Pauw device fabricated on a common gate and insulator. All contacts are projected on the same plane for convenience. (**c**) Potential map and current density streamlines in the van der Pauw device in linear regime. The scale bar is 0.5 mm long. The simulation is parameterized with a current $I_1 = 1\,\mu A$ flowing through a film with sheet conductance $\sigma_s = 1.5\,\mu S\,sq^{-1}$. The dashed lines are equipotential lines $V_3, V_4$ and $V_C = (V_3 + V_4)/2$.

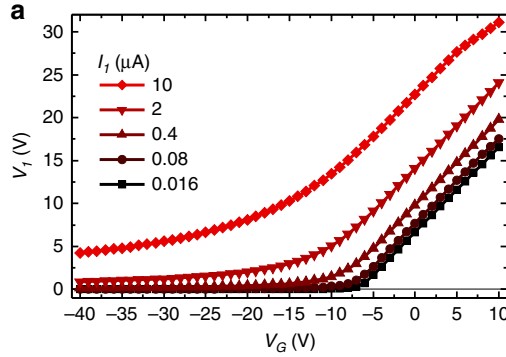

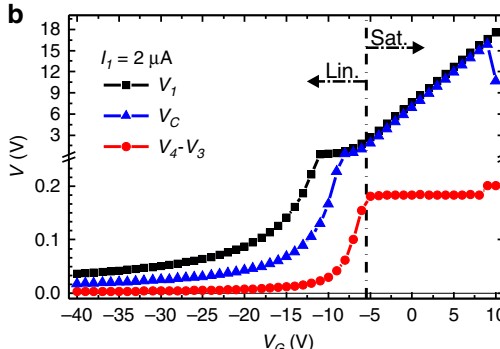

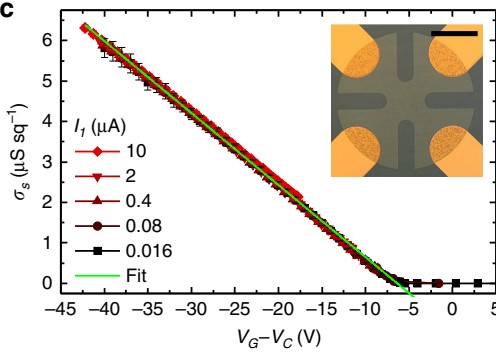

**Figure 2 | Characterization of a gVDP device.** (**a**) Potential $V_1$ of the injecting contact 1 as a function of gate voltage $V_G$ for different currents $I_1$. (**b**) Evolution with $V_G$ of three characteristic potentials $V_1, V_C = (V_3 + V_4)/2$ and $V_4-V_3$. Data measured with $I_1 = 2\,\mu\text{A}$. (**c**) Sheet conductance $\sigma_s$ of the semiconductor film extracted using the gVDP method at different $I_1$. $\sigma_s$ is plotted as a function of $V_G$-$V_C$. The error bars are computed by averaging over 8 measurements, 2 along each side of the gVDP structure. The line is linear fit. Inset: photograph of the gVDP device based on a thin film of $C_{10}$-DNTT shaped as a clover-leaf. Scale bar is 0.5 mm long.

convenience. Applying a potential $V_G$ to the gate relative to the grounded contact 2 leads to the accumulation of charges at the semiconductor/insulator interface. This results in an increase of $\sigma_s$ that promotes current flow. The relationship between $\sigma_s$ and $V_G$ is explained by a model derived from the TFT generic charge drift model given by:[23]

$$I_D \frac{L}{W} = \sigma_s (V_D - V_S)$$
$$= \mu_{tfsc} C_I \frac{(V_G - V_T - V_S)^{\gamma+2} - (V_G - V_T - V_D)^{\gamma+2}}{\gamma + 2}, \quad (1)$$

where $I_D$ is the TFT drain current, $L$ and $W$ are the channel length and width, $V_S$ and $V_D$ are the potential at the source and

drain contacts, $C_I$ is the gate insulator capacitance per unit area and $\gamma$ is the mobility enhancement factor. To model gVDP operation with equation 1, we simply treat the probed region in the VDP device as a TFT with source and drain at potentials $V_4$ and $V_3$ respectively, and with geometrical dimensions $L/W = \ln(2)/\pi$, as demonstrated by van der Pauw for a square VDP structure[18]. Assuming that $\mu_{tfsc}$ is unaffected by potential variation, that is $\gamma = 0$, the generic TFT model is rewritten as:

$$I_1 \frac{\ln(2)}{\pi} = \sigma_s |V_3 - V_4|$$
$$= \frac{\mu_{tfsc} C_I}{2} \left[ (V_G - V_T - V_4)^2 - (V_G - V_T - V_3)^2 \right], \quad (2)$$

and after simplification:

$$\sigma_s = \mu_{tfsc} C_I |V_G - V_T - V_C|, \quad (3)$$

with $V_C = \frac{V_3 + V_4}{2}$. In the parallel plate capacitor formed by the gate/insulator/semiconductor stack, the charge density is $\delta = C_I |V_G - V_T - V_C|$, where $V_C$ approximates the potential in the probed region of the gVDP device. Equation 3 is very similar to the gradual channel approximation model of a TFT in the linear regime that is derived from equation 1 with $V_S = 0$ and $\gamma = 0$:

$$\sigma_s = \frac{I_D}{V_D} \cdot \frac{L}{W} = \mu_{app} C_I \left| V_G - V_T - \frac{V_D}{2} \right|. \quad (4)$$

Here, $\frac{V_D}{2}$ approximates the potential in the TFT channel and $\mu_{app}$ is the apparent mobility, which, in contrast to $\mu_{tfsc}$, is affected by the contact resistance.

**Gated van der Pauw device operation.** All devices fabricated in this study were based on the bottom gate-top contact staggered topology depicted in Fig. 1b. Substrates were $2 \times 2$ cm highly doped Si wafers acting as common gate with $\sim 125$ nm thermally grown $SiO_2$ as insulator. After a surface treatment with self-assembled monolayers, we thermally evaporated thin ($\leq 30$ nm) films of organic semiconductors through a shadowmask. Then we evaporated metallic electrodes using a second aligned shadowmask. For gVDP devices, we favored clover leaf patterns that allow easy-to-apply large contacts and are tolerant against misalignment. Also, TFT devices with various channel lengths were processed simultaneously. They were used to generate transmission line measurements (TLM) from which reference $\mu_{app}$ and $R_c$ were extracted.

gVDP electrical characterization requires a current source to control $I_1$. The bias on contact 1, $V_1$, is automatically adjusted to keep a constant current, while sweeping the gate voltage $V_G$. Furthermore, the gVDP device is best measured when the grounded electrode 2 is also the contact that sinks charge carriers, as shown in Fig. 1a. Therefore, for p-type (n-type) semiconductors, holes (electrons) are injected by the positively (negatively) biased contact 1, resulting in a positive (negative) current $I_1$ flowing from 1 to 2. In the case of p-type organic semiconductor $C_{10}$-DNTT (2,9-didecyl-dinaphtho-[2,3-b:20, 30;-f]-thieno-[3,2-b]-thiophene) with gold contacts, Fig. 2a shows the evolution of $V_1$ as a function of $V_G$ for a broad series of $I_1$. We distinguish two operating regimes from the shape of the $V_1$ curves. At high (positive) $V_G$, $V_1$ has a linear dependence with $V_G$ with a slope equal to unity. At low (negative) $V_G$, $|V_1|$ is small and slowly decreases as $|V_G|$ increases. These regimes are respectively called saturation and linear regimes, as they correspond to the eponymous regimes observed in TFT operation.

The two regimes are also apparent in Fig. 2b, where $V_1$, $V_C = \frac{V_3 + V_4}{2}$ and $V_4-V_3$ are detailed for $I_1 = 2\,\mu\text{A}$. In the saturation regime, $V_C$ follows $V_1$ very closely, while $V_4-V_3$ takes

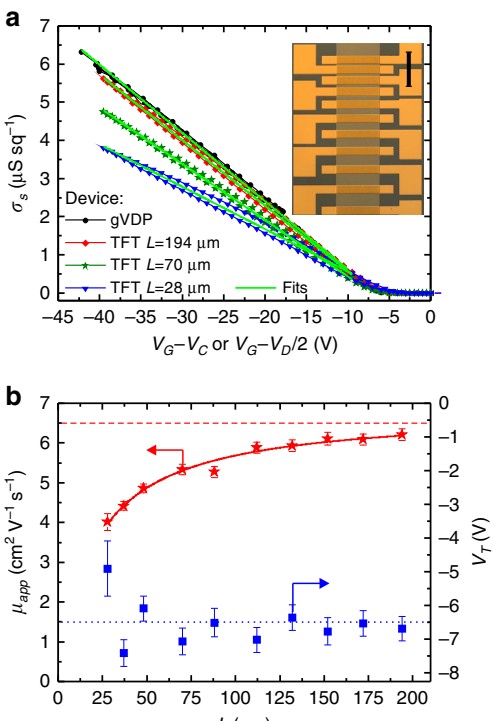

**Figure 3 | Comparison with TFT characteristics.** (**a**) Sheet conductance $\sigma_s$ of the semiconductor thin film measured from a gVDP device and three TFTs with different channel lengths taken from a TLM device. Lines are linear fits delivering the apparent mobility $\mu_{app}$ and the threshold voltage $V_T$ of the TFTs. Inset: photograph of the TLM structure with a thin $C_{10}$-DNTT film and Au top contacts. Scale bar is 0.5 mm. (**b**) Evolution of $\mu_{app}$ and $V_T$ with TFT channel length. $\mu_{app}$, $V_T$ and the error bars are obtained from the linear regression on the $\sigma_s$ versus $V_G - V_D/2$ characteristic of each TFT, as shown for three values of $L$ in (**a**). The line is a fit to the $\mu_{app}$ data. The dashed and dotted horizontal lines respectively represent the $\mu_{tfsc}$ and $V_T$ from the corresponding gVDP device.

a small constant value of 0.19 V. In saturation, the majority of the semiconductor film is subjected to a limited potential drop. Its average potential $V_C$ is close to $V_1$. In this almost equipotential region, $V_1$ (hence $V_C$) automatically adjusts to a value substantially higher than $V_G$ so that the charge density $\delta_{sat} = C_I |V_G - V_T - V_C|$ remains constant at a value allowing current flow. In contrast, the vicinity of grounded contact 2 is depleted of charge carriers. A potential drop through the semiconductor film, from $\sim V_C$ to $V_2 = 0$ V, creates a lateral field sufficient to maintain current through this depletion zone. In the linear regime, Fig. 2b shows that $V_C$ is exactly the half of the small $V_1$. In this case, as represented in the simulation of Fig. 1c, the potential linearly drops when current flows from contact 1 to 2. The transition between regimes is most visible in the $V_4 - V_3$ curve in Fig. 2b. It takes place at $V_G \sim V_T$ corresponding to the closure of the depletion zone around contact 2. Further $V_G$ decrease into the linear region yields an increase in charge carrier density so that $\delta_{lin} > \delta_{sat} = cst$. The increase of $\delta_{lin}$ is compensated by the progressive lowering of the lateral electric field (seen in the decrease of $V_1$ and $V_4 - V_3$). This maintains a constant current $I_1$ throughout the linear regime.

We repeated the measurement detailed in Fig. 2b on both directions of all four sides of the $C_{10}$-DNTT gVDP device shown in the inset of Fig. 2c. After averaging the eight measurements, the sheet conductance of the semiconductor film $\sigma_s$ is extracted using the VDP method for various $V_G$ and $I_1$. Following equation 3, $\sigma_s$ is plotted as a function of $V_G - V_C$ in Fig. 2c. With this choice of

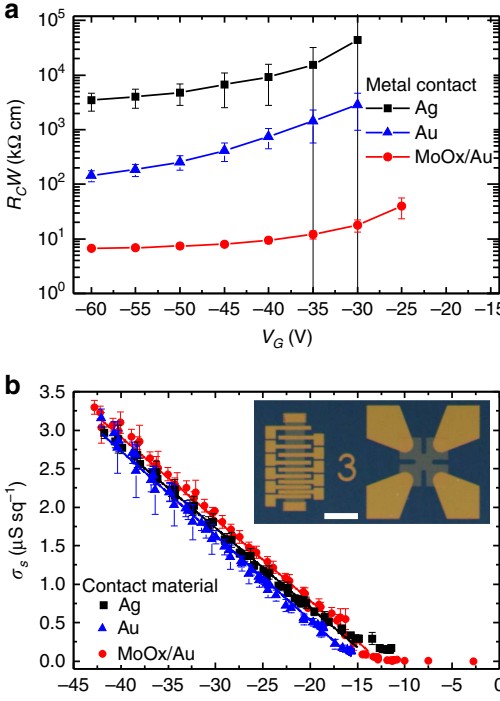

**Figure 4 | Effect of contact resistance on gVDP measurements.** (**a**) $V_G$ dependence of contact resistance $R_c$ extracted from TLM measurements. The error bars are obtained from the linear regression over the total resistance versus channel length data that yields $R_c$. The TLM devices are based on $C_8$-BTBT films with three different top contact materials yielding large differences in $R_c$. (**b**) $\sigma_s$ is extracted by gVDP from the same $C_8$-BTBT films with different contacts, at currents $I_1$ ranging from 0.04 to 5 μA. The overlap of the $\sigma_s$ versus $V_G - V_C$ plots show the independence of the gVDP method from $R_c$. The error bars are computed by averaging over 8 measurements, 2 along each side of the gVDP structure. Lines are linear fits. Inset: photograph of a sample with side-by-side TLM and gVDP devices. Scale bar is 1 mm long.

the X-axis, all characteristics fall along the same straight line. In contrast, Supplementary Fig. 1 shows the $\sigma_s$ versus $V_G$ plot that does not lead to any useful interpretation. The line in Fig. 2c is a linear fit using equation 3. Its slope and intercept with the X-axis give $\mu_{tfsc} = 6.5 \pm 0.1 \; \text{cm}^2 \, \text{V}^{-1} \, \text{s}^{-1}$ and $V_T = -6.5 \pm 0.2$ V, respectively. The straightness of the curve in Fig. 2c confirms that $\mu_{tfsc}$ has negligible field dependence ($\gamma = 0$). Also the small error bars in Fig. 2c, and the limited s.d. in the extracted data show that the gVDP approach effectively reduces error and increases precision. Finally, the two regimes observed earlier, do not appear in the evolution of $\sigma_s$ with $V_G - V_C$. Indeed, $\sigma_s$ is extracted from the probed region between equipotentials $V_3$ and $V_4$ drawn in Fig. 1a. Since this region remains far from the depletion zone in the vicinity of grounded contact 2, it always experiences a linear current transport, even when the gVDP device is driven in saturation. In consequence, the extraction of $\sigma_s$ is independent of the regime of operation and data can be collected across a broad $V_G$ range.

**Comparison with TFT measurements.** In parallel to the gVDP device discussed so far, we prepared a TLM device with ten TFTs based on the same $C_{10}$-DNTT film. The TFTs had a channel width $W = 630$ μm and channel lengths $L$ ranging from 28 to 194 μm (Supplementary Fig. 2 shows some TFT transfer and

**Table 1 | Electrical characteristics of thin evaporated C$_8$-BTBT films with three different contact materials.**

| Contact material | $R_c$ (kΩ cm) TLM $V_G = -60$ V | $\mu_{app}$ (cm$^2$V$^{-1}$s$^{-1}$) TFT Sat $V_D = -60$ V | $\mu_{app}$ (cm$^2$V$^{-1}$s$^{-1}$) TFT Lin $V_D = -1$ V | $\mu_{tfsc}$ (cm$^2$V$^{-1}$s$^{-1}$) gVDP | $V_T$ (V) TFT Sat $V_D = -60$ V | $V_T$ (V) TFT Lin $V_D = -1$ V | $V_T$ (V) gVDP |
|---|---|---|---|---|---|---|---|
| MoOx/Au | 6.67 ± 0.29 | 4.1 ± 1.6 | 2.6 ± 0.5 × 10$^{-1}$ | 3.9 ± 0.1 | −16.9 ± 5.9 | −14.9 ± 2.5 | −12.6 ± 0.7 |
| Au | 145 ± 32 | 3.6 ± 1.2 | 4.6 ± 1.4 × 10$^{-2}$ | 4.0 ± 0.1 | −37.2 ± 4.1 | −43.0 ± 5.2 | −15.1 ± 0.3 |
| Ag | 3440 ± 1200 | 4.5 ± 0.8 | 4.5 ± 6.0 × 10$^{-4}$ | 3.8 ± 0.1 | −42.4 ± 1.3 | −29.0 ± 6.9 | −13.2 ± 0.4 |

All devices, TLM, TFT ($W/L = 630/28$ μm) and gVDP are fabricated on the same sample.

output characteristics). Following equation 4, values of $\sigma_s$ from three TFTs with different $L$ are plotted in Fig. 3a as a function of $V_G$-$V_D$/2, along with the $\sigma_s$ obtained from the gVDP device. The TFT curves in Fig. 3a show a slight hysteresis with the back and forth sweep of $V_G$, especially for the short channel devices. gVDP devices usually display the same level of hysteresis as long channel TFTs. In the case of the C$_{10}$DNTT gVDP device discussed so far, this hysteresis is negligible.

The lines in Fig. 3a are linear fits using equation 4 that deliver the apparent mobility $\mu_{app}$ and threshold voltage $V_T$ of the TFTs. We used this approach to extract $\mu_{app}$ and $V_T$ of all 10 TFTs in the TLM structure. These are reported in Fig. 3b as a function of $L$. As $L$ increases, the TFT sheet conductance curve in Fig. 3a shifts towards the gVDP curve. This progression also appears in Fig. 3b, where $\mu_{app}$ tends towards $\mu_{tfsc}$ with the increase of $L$. This can be formalized using the following relation between $\mu_{app}$ and $\mu_{tfsc}$[24,25]:

$$\mu_{app} = \frac{\mu_{tfsc}}{1 + \frac{L_{1/2}}{L}}, \qquad (5)$$

where $L_{1/2}$ is the channel length at which the contact resistance $R_c$ is equal to the channel resistance $R_{ch}$. Fitting equation 5 to $\mu_{app}$ in Fig. 3b delivers $\mu_{tfsc} = 6.8 \pm 0.4$ cm$^2$V$^{-1}$s$^{-1}$ and a $L_{1/2} = 20 \pm 1$ μm. This value of $\mu_{tfsc}$ is within range of the gVDP value (dashed line in Fig. 3b). The non-negligible $L_{1/2}$ shows that $R_c$ affects all TFTs in the TLM structure, which explains the performance degradation with the shrinking of $L$. In contrast, $L$ variation does not affect the threshold voltage $V_T$ except for some scattering at low $L$ caused by non-linearities in the $\sigma_s$ curves (Fig. 3b). On average, the $V_T$ extracted from TFT measurements stays close to the value measured by gVDP (dotted line in Fig. 3b).

We now apply the standard TLM analysis. The variation of the total device resistance $R_{tot}$ with $L$ can be expressed as:[13,24,25]

$$R_{tot} W = R_c W + \frac{L}{\mu_{tfsc} C_I |V_G - V_T|}. \qquad (6)$$

This procedure yields $\mu_{tfsc}$ from the slope of the fit and $R_c W$ from its intercept with the $Y$-axis. Supplementary Figure 3 shows the fitting procedure and the variation of the extracted parameters with $V_G$. At $V_G = -40$ V, we obtain $\mu_{tfsc} = 6.8 \pm 0.2$ cm$^2$V$^{-1}$s$^{-1}$ and $R_c W = 330 \pm 30$ Ω cm. This $\mu_{tfsc}$ exactly matches the value obtained from the fitting of $\mu_{app}$ in Fig. 3b. It is also within the error of the $\mu_{tfsc}$ measured in the gVDP device. $R_c W$ has a low value for organic TFTs that is characteristic of this material set[14,25]. TFT characteristics are nevertheless still seriously impacted by $R_c$ because of the high-mobility, hence low $R_{ch}$, of the C$_{10}$-DNTT film[14]. The gVDP device, on the other hand, shows an ideal behaviour and its characteristic in Fig. 3a represents the optimum towards which TFTs tend as the effect of $R_c$ abates.

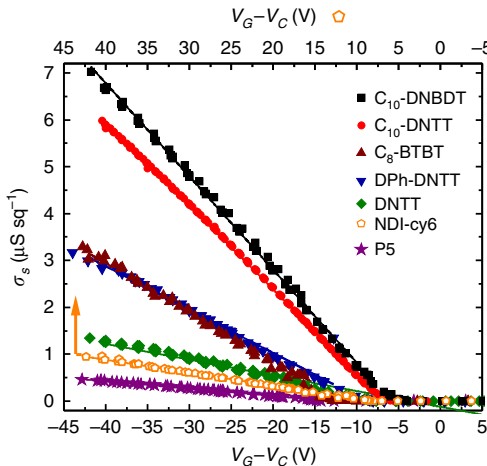

**Figure 5 | gVDP characteristics of thin films of seven different organic semiconductors.** $V_G$-$V_C$ dependence of the sheet conductance $\sigma_s$ measured by gVDP on evaporated films of six p-type semiconductors and one n-type semiconductor (NDI-cy6, upper X scale). $I_1$ ranges from 0.04 to 10 μA. The lines are linear fits for data extraction.

**Influence of contact resistance**. The gVDP device studied so far has large dimensions. The electrical current path length is >1 mm, that is much longer than $L_{1/2}$. This points to the fact that $R_c$ in this device is negligible compared to its total resistance $R_{tot}$. Indeed, at $V_G = -40$ V, $R_{tot} = V_1/I_1 \sim 400$ kΩ and we estimate $R_c \sim 4$ kΩ from the dimensions of the contact pads. Our C$_{10}$-DNTT gVDP device is hardly sensitive to contact effects and we cannot draw conclusions regarding the impact of more serious $R_c$ on gVDP device operation. To examine this question, we fabricated gVDP and TLM devices based on evaporated thin films of C$_8$-BTBT (2,7-dioctyl-[1]benzothieno-[3,2-b][1]benzothiophene) that is more prone to contact problems. Indeed, C$_8$-BTBT has a deep HOMO level of −5.8 eV that complicates band alignment at the metal/semiconductor interface and poor vertical transport properties that complicate access to the channel in the staggered device topology[26,27]. Intercalating a thin layer of MoOx has been previously shown to improve injection[28]. In consequence, the three different electrode materials tested here, MoOx/Au, Au and Ag, yield a significant variation in $R_c$, as can be seen from the TLM analysis in Fig. 4a. Supplementary Figure 4 shows the TFT transfer characteristics measured in linear and saturation regimes of the shortest channel devices ($W/L = 630/28$ μm). The TFT with the least resistive contacts, MoOx/Au, is already affected by $R_c$ as its $I_D$ and $\sqrt{I_D}$ curves show substantial sublinear behaviour. Using worse contact materials completely depresses current in the linear regime and delays threshold in the saturation regime. Table 1 reports values of $\mu_{app}$ and $V_T$ of these short channel TFTs extracted using the

**Table 2 | Electrical characteristics of thin evaporated films of 7 different organic materials.**

| Organic semiconductor | $R_c$ (k$\Omega$ cm) TLM | $\mu_{app,sat}$ (cm$^2$V$^{-1}$s$^{-1}$) TFT Sat | $\mu_{app,lin}$ (cm$^2$V$^{-1}$s$^{-1}$) TFT Lin | $\mu_{tfsc}$ (cm$^2$V$^{-1}$s$^{-1}$) gVDP | $V_T$ (V) TFT Sat | $V_T$ (V) TFT Lin | $V_T$ (V) gVDP |
|---|---|---|---|---|---|---|---|
| C$_{10}$-DNBDT | 0.85 ± 0.06 | 7.3 ± 1.7 | 5.6 ± 0.6 | 7.2 ± 0.1 | − 3.5 ± 0.5 | − 3.2 ± 1.7 | − 5.9 ± 0.2 |
| C$_{10}$-DNTT | 0.33 ± 0.03 | 7.5 ± 0.3 | 6.1 ± 0.3 | 6.5 ± 0.1 | − 7.0 ± 0.4 | − 6.4 ± 1.0 | − 6.5 ± 0.1 |
| C$_8$-BTBT | 6.67 ± 0.29 | 3.9 ± 0.4 | 1.4 ± 0.5 | 3.9 ± 0.1 | − 22.1 ± 0.9 | − 20.4 ± 1.2 | − 12.6 ± 0.7 |
| DPh-DNTT | 0.92 ± 0.06 | 3.7 ± 0.3 | 3.3 ± 0.4 | 3.3 ± 0.1 | − 6.0 ± 0.6 | − 7.9 ± 1.9 | − 8.9 ± 0.7 |
| DNTT | 1.35 ± 0.20 | 1.4 ± 0.2 | 1.2 ± 0.2 | 1.2 ± 0.1 | − 3.6 ± 1.0 | − 2.3 ± 1.4 | − 3.6 ± 0.5 |
| NDI-cy6 | 28.8 ± 3.8 | 1.1 ± 0.2 | 0.44 ± 0.10 | 1.4 ± 0.1 | 13.3 ± 1.0 | 10.2 ± 1.0 | 8.0 ± 0.2 |
| Pentacene | 13.8 ± 0.5 | 0.59 ± 0.31 | 0.37 ± 0.06 | 0.56 ± 0.05 | − 11.2 ± 1.8 | − 11.7 ± 3.0 | − 12.9 ± 0.2 |

All devices, TLM, TFT ($W/L = 630/194\,\mu$m) and gVDP are fabricated on the same sample. $R_c$ is given at the highest measured $|V_G|$. At least three TFTs per sample are measured in saturation and linear regimes.

gradual channel approximation. The data show important spread reflecting the low-quality of the transfer curves. In particular, $V_T$ is strongly affected by contact quality, especially in short channel devices. This is caused by a voltage offset set by the non-ideal contact between the reference injecting electrode and the transistor channel that is not taken into account in the gradual channel approximation model used for $V_T$ extraction[29,30]

Next, we measured gVDP devices fabricated on the same three samples with $I_1$ ranging from 0.04 to 5 µA. We estimate from the $R_cW$ value at $V_G = -60$ V that the worst contact material, Ag, yields an estimate $R_c \sim 40$ MΩ in the C$_8$-BTBT gVDP device. Such $R_c$ is far superior to the expected film resistance in the on region and dominates gVDP device operation. In spite of this, the $\sigma_s$ characteristics of the three gVDP devices with different contact materials are linear and superimpose in Fig. 4b. The values of $\mu_{tfsc}$ reported in Table 1 have little spread and are all within error of each other. The values of $V_T$, also in Table 1, show a slight spread that may have the same origin as the heavy $V_T$ spread seen above in TFTs, but of a much lower magnitude since the reference electrode is not the injecting one. The independence of $\mu_{tfsc}$ from the nature of the contact material and the low spread in $V_T$ demonstrates that the gVDP method is quite insensitive to $R_c$, even when charge injection completely dominates transport as in the case of Ag contacts on C$_8$-BTBT. In gVDP devices, the region probed by sensing contacts 3 and 4 is not influenced by current injection and extraction in contact 1 and 2, respectively.

**Other semiconductors.** Besides C$_{10}$-DNTT and C$_8$-BTBT, we have investigated thin films of five other evaporated organic semiconductors, namely p-type C$_{10}$-DNBDT (3,11-didecyl-dinaphtho-[2,3-d:20,30-d0]-benzo-[1,2-b:4,5-b0]-dithiophene), DPh-DNTT (2,9-diphenyl-dinaphtho-[2,3-b:20,30;-f]-thieno-[3,2-b]-thiophene), DNTT (dinaphtho-[2,3-b:20,30;-f]-thieno-[3,2-b]-thiophene), pentacene and n-type NDI-cy6 (2,7-Dicyclohexylbenzo [lmn] [3,8] phenanthroline − 1,3,6,8(2H,7H)–tetraone), also known as DCyNTDA. Thin (∼ 30 nm) films of these materials were deposited in vacuum using growth conditions optimized for electrical performance. Characterization by X-ray diffraction and atomic force microscopy is given for all films in Supplementary Fig. 5 and Supplementary Table 1. As discussed in Supplementary Note 1, this characterization shows that all thin films are polycrystalline with growth patterns that are beneficial to lateral charge transport: the films are composed of a mosaic of two-dimensional grains with diameters ranging from 0.5 to > 5 µm, that is much smaller than lateral gVDP device dimensions. These grains present a layer-by-layer microstructure with molecules standing on their long axis, which maximizes electronic coupling between adjacent molecular cores within the plane of the layer. In most cases, this continuous two-dimensional film is covered by a dense matrix of tall-elongated

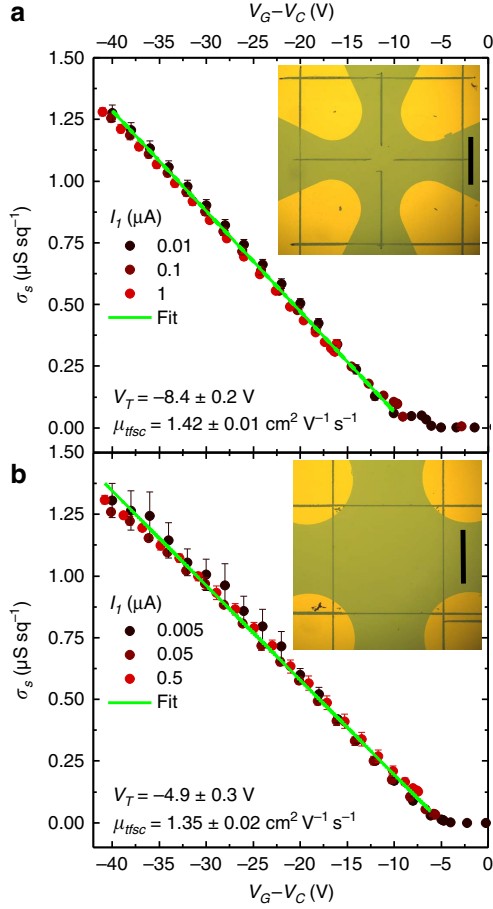

**Figure 6 | gVDP devices based on organic films patterned by scratching.** The samples consist of thin (∼ 30 nm) DNTT films evaporated on common gate Si/SiO$_2$ substrates, without patterning. Au contacts are then patterned by evaporation through a shadowmask. No mask alignment is necessary. Upon electrical characterization, a probe needle is used to manually pattern the DNTT film by scratching it. (**a**) Scratched clover-leaf pattern. (**b**) Scratched square pattern. In both figures, the error bars are computed by averaging over 8 measurements, 2 along each side of the gVDP structure. The insets show photographs of the scratched devices. Scale bars are 1 mm long. Both devices were produced more than three months apart, which may explain the difference in $V_T$.

needles. This three-dimensional growth is symptomatic of a Stranski–Krastanov roughening transition.

For all semiconductors, gVDP measurements deliver well-behaved $\sigma_s$ characteristics (see Fig. 5) with a linear behaviour

**Table 3 | Fabrication conditions of electrical devices based on thin films of different organic semiconductors.**

| Organic semiconductor | Material supplier | Thermal gradient purification | Self-assembled monolayer | Deposition rate ($Å s^{-1}$) | Substrate temperature (°C) | Contact material |
|---|---|---|---|---|---|---|
| $C_{10}$-DNBDT | Pi-Crystal Inc. | $0 \times$ | ODTS | 0.20 | 135 | Au |
| $C_{10}$-DNTT | Nippon Kayaku Co. | $0 \times$ | ODTS | 0.10 | 80 | Au |
| $C_8$-BTBT | Nippon Kayaku Co. | $1 \times$ | PETS | 0.15 | 25 | MoOx/Au* |
| DPh-DNTT | Nippon Kayaku Co. | $0 \times$ | ODTS | 0.15 | 70 | Au |
| DNTT | Nippon Kayaku Co. | $1 \times$ | ODTS | 0.50 | 75 | MoOx/Au |
| NDI-cy6 | Lumtec Co. | $1 \times$ | TDPA | 0.30 | 90 | Ca/Ag |
| Pentacene | Sigma-Aldrich Co. | $1 \times$ | ODTS | 0.25 | 65 | Au |

*Au and Ag were also used to form lower quality contacts on $C_8$-BTBT.

confirming the field-independence of the mobility. Data extraction using equation 3 is straightforward and $\mu_{tfsc}$ and $V_T$ are reported for each material in Table 2. These values are compared with TFTs ($W/L = 630/194\,\mu m$) measured on the same sample and analysed using the gradual channel approximation model. Even with long channel lengths, many TFT transfer curves show non-linearity that induce serious variations of $\mu_{app}$ and $V_T$ with $V_G$ and yield large s.d. over the TFT data in Table 2. The mobility of TFTs in the linear regime remains inferior to the thin film mobility obtained by gVDP, $\mu_{app,lin} \leq \mu_{tfsc}$. This difference increases as the importance of $R_c$ relative to $R_{tot}$ increases. On the other hand, in cases where $R_c$ is not too high, the TFT mobility in the saturation regime $\mu_{app,sat}$ is slightly superior to $\mu_{tfsc}$. The $V_T$ from TFTs and gVDP are similar for most cases, except for material systems such as $C_8$-BTBT with MoOx/Au, where an important interfacial energy barrier imposes a potential drop to charge the channel, resulting in an increase of the apparent $|V_T|$ of the TFT. In conclusion, contact non-idealities complicate TFT data analysis: non-linearity of the transfer characteristics compromises the quality of the extracted values, which depend on channel length and measurement regime. Such problems are absent in the gVDP method: Characteristics are linear, independent from device dimensions and measurement regime. They deliver trustworthy data with low-spread. In consequence, the gVDP method is an excellent probe to systematically relate electrical performance with the morphology and microstructure of thin films of organic semiconductors. Such systematic growth studies are left to further work.

## Discussion

Our motivation to develop the gVDP method is a simple and accurate extraction of the mobility of thin semiconductor films that is representative of TFT operation. Other methods for contact-independent mobility extraction in the high-charge density regime exist such as Hall-effect measurements[31–33] and field-induced time-resolved microwave conductivity[1,34]. These techniques, however, involve specialized measurement setups that are not broadly accessible. Other approaches exist to get $\mu_{tfsc}$ from electrical measurements only. In the Results section, we have employed two methods to obtain $\mu_{tfsc}$ from TLM data. Although accurate, TLM requires multiple device measurements and mobility extraction is weakened by the choice of $V_T$ that parameterizes equation 6 (refs 25,29,35). Methods involving advanced device modelling of TFT characteristics also exist, although they require setting up complex measurements and/or data treatment schemes[35–38].

Besides, $\mu_{tfsc}$ can be readily obtained from the measurement of a gated Four Point-Probe (gFPP) device, where the functions of current injection and voltage measurement are separated in the

channel[39,40]. The gFPP device fabrication is however complicated by the precise alignment of the voltage probes along the very edge of the semiconductor channel. This requires advanced patterning techniques and small variations in device geometry can compromise results[41]. In contrast, the gVDP device is much simpler to fabricate: using a clover-leaf pattern greatly simplifies alignment down to small device size. Patterning of the organic layer is not even necessary: from a continuous DNTT film with patterned Au contacts, we could roughly shape square and clover-leaf patterns with contacts at the corners by a simple scratching with a probe needle and still obtain excellent gVDP measurements (Fig. 6). Measuring the gVDP device is of the same complexity as the gFPP device measurement: Both require five contacts. The data obtained are, however, more precise than in gFPP thanks to the averaging over all sides and the independence from geometric dimensions.

All films analysed in this study were polycrystalline with random grain orientation and grain size much smaller than device dimensions. In consequence, they all displayed isotropic transport properties as evidenced by equal resistances $R_{12}$ and $R_{23}$ measured along two perpendicular edges of gVDP devices with four-fold symmetry. The gVDP method could, however, be extended to the treatment of anisotropic thin film of semiconductors such as thin organic single crystals by adapting methods previously developed for the interpretation of VDP measurements carried out on anisotropic films[42,43].

Besides the small molecular organic semiconductors studied here, the gVDP method is generally suited to characterize a wide range of materials, such as semiconducting polymers, metal oxides, 2D materials and so on. The characterization of very low-mobility semiconductors is ultimately limited by the resolution of the measurement setup. Strategies to enhance current such as device downscaling and the use of stronger dielectrics can help these measurements. In the case of semiconductor/dielectric systems that display significant field-dependence of the mobility ($\gamma > 0$), a superlinear behaviour of the $\sigma_s$ versus ($V_G - V_C$) characteristic is expected. Data analysis would require the derivation of a model equivalent to equation 3 that still contains the mobility enhancement factor $\gamma$.

In conclusion, we develop the gated van der Pauw (gVDP) method for the electrical characterization of thin semiconducting films. This method combines the following advantages: (1) Device structure and fabrication constraints are the same as for thin film transistors, allowing easy device integration and comparison. (2) Independence from contact effects that are detrimental to transistor characteristics. (3) Straightforward data analysis using equation 3 and precise parameter extraction thanks to the inherent averaging and independence from geometrical dimensions. We tested this method on thin films of seven high-mobility organic semiconductors of both polarities, but it is applicable to any other thin film semiconductor. We show that the

gVDP method delivers accurate values for the charge carrier mobility and the threshold voltage of these films in the high-charge density accumulation layer that is representative of transistor operation. Finally, this method is inherently independent from contact effects as the probed region is remote from metal electrodes. The gVDP device is therefore an excellent probe to systematically relate electrical characteristics to the morphology and microstructure of the thin film semiconductor. It is also a great vehicle for physical studies that combine electrical measurements with other excitations, for example, magnetic field or light.

## Methods

**Device fabrication.** All devices were fabricated on $SiO_2/Si$ $n^{++}$ substrates, the thickness of $SiO_2$ was 125 nm. In the case of the NDI-cy6, an additional 100 nm of dielectric $Al_2O_3$ was grown by atomic layer deposition on top of the $SiO_2$. All substrates were cleaned with solvents and exposed to ultraviolet-ozone for 15′, followed by a treatment with a self-assembled monolayer, as detailed in Table 3. Octadecyl-trichloro-silane (ODTS) and phenyl-ethyl-trichloro-silane (PETS) treatments were applied to the $SiO_2$ surface by exposing the substrate to vapour of the liquid precursor at 140 °C in a vacuum chamber for 1 h. n-Tetradecyl-phosphonic acid (TDPA) treatment was applied to the surface of $Al_2O_3$ by immersing the substrate in a solution of TDPA:2-propanol $5 \times 10^{-3}$ M for 19 h. Thin films ($\sim 30$ nm) of organic semiconductors were evaporated in high vacuum ($1 \times 10^{-8}$ torr) through a shadow mask, using optimized conditions given in Table 3. This table also informs on the material suppliers and the thermal gradient purification used after receiving the material. Electrodes were vacuum deposited through a manually aligned shadow mask and with a substrate temperature of $\sim -5$ °C. 5 nm of MoOx, 40–60 nm of Au, 20 nm of Ca and 60 nm of Ag, were deposited at rates of 0.05, 1, 1 and 2 Å s$^{-1}$, respectively. The following procedure was used for the patterning of organic semiconductor films by scratching shown in Fig. 6: the sample was mounted and aligned on the probe station. A probe needle was put in contact with the sample, and the sample stage was moved to produce straight scratching lines parallel to edges of the Au patterns. Points of attention when shaping the gVDP device: (1) Contact size remains small in comparison with semiconductor pattern size: the radius of contacts in the square devices must be at least 10 times smaller than the square side length. The clover-leaf pattern provides much more relaxed size constraints. (2) Devices keep a four-fold symmetry. Thanks to the inherent averaging, the extracted value remains accurate, but the error increases in case of asymmetry.

**Device characterization.** All electrical measurements were performed either in air or in a $N_2$ glovebox, using a probe station connected to an HP4156C parameter analyser. For gVDP device measurement, Contacts *1,2* and the gate were each connected to a source monitor unit, contacts *3* and *4* were each connected to a voltage measurement unit. The parameter analyser was remote controlled by a Labview program that piloted both measurements along each side of the device. It is worth noting that a large potential difference between contact *1* ($V_1$) and the gate ($V_G$) in the saturation regime can lead to bias stress effects resulting in a negative (positive) $V_T$ shift for p-type (n-type) semiconductors. This is avoided by setting a low voltage compliance on $V_1$ (we used $+40$ V) and by avoiding too large $I_1$. The maximum $I_1$ that can be sustained by a sample depends mainly on the semiconductor mobility, the gate capacitance and the length and width of the current path between contacts *1* and *2*. For organic semiconductors, the typical range of $I_1$ is between 10 nA and 10 µA. Throughout the measurements we always measured gate current $I_G$ to monitor gate leakage. In conventional TFT analysis, the values of $\mu_{app}$ were estimated by the conventional gradual channel approximation model given by equation 4 in the linear regime and $\mu_{app,sat} = (2L/W)(1/C_I)(\partial \sqrt{I_D}/\partial V_G)^2$ in the saturation regime.

**Data availability.** The data that support the findings of this study are available from the corresponding author on reasonable request.

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

## Acknowledgements

This work has received funding from the European Research Council under the European Union's Seventh Framework Programme (FP7/2007-2013)/ERC grant agreement no. 320680 (EPOS CRYSTALLI) and from the Research Foundation Flanders (FWO Vlaanderen) under the FWO-ARRS research collaboration program/grant number G0B5914N (ORSIC-HIMA). We thank Nippon Kayaku Co. for supplying the C$_8$-BTBT, DNTT, C$_{10}$-DNTT and DPh-DNTT used in this study.

## Author contributions

C.R. and G.B. conceived the original idea. J.G. and C.R. derived the theoretical framework. E.K., J.-H.L. and C.R. fabricated devices. E.K. and C.R. performed the electrical characterization of the devices. J.G., G.B. and P.H. oversaw experiments. C.R. prepared the manuscript. All authors discussed the results and revised the manuscript.

## Additional information

**Competing interests:** The authors declare no competing financial interests.

