## [Peer Review File · Nature Communications]

Reviewers' comments:

Reviewer #1 (Remarks to the Author):

Probing the intrinsic charge carrier mobility in thin films of organic semiconductors by the gated van der Pauw method

At first, I would like to commend the authors their successful evaluation of mobility in this simple but precise analysis technique, van der Pauw method giving the real mobility of charge carriers mostly free from electrical unknown parameters originated from device structures. Especially the mobility derived from the demonstrated devices depends slightly on the contact resistances varying more than two orders of magnitude, suggesting extremely high feasibility of the present technique. Thus I am principally very positive to support publication of this manuscript in Nature Commun.

Only my concerns are,

1) Definitely the value of mobility has been evaluated after careful elimination of the effects from contact resistance, etc., and on this point, I am much convinced that the value of mobility is more reliable as the "real" values than those derived from the simple TFT device configurations. However still the value is estimated eventually as the results of long-range translational motion of charge carriers among the electrodes, and even with this unique Clover-leaf like configuration of the electrodes, the range of translational motion is in the order of microns which is large enough in relation to the grain size. Thus I feel it inadequate to refer the mobility as "intrinsic".

2) Please give some parameter on the solid state structure of materials deposited on the substrate such as averaged grain sizes, crystallite orientations, etc. by microscopic observation and/or XRD, otherwise the readers cannot compare the mobility values to the others.

3) In case of Ag contact, extremely high contact resistance was observed as 40 M Ω , but why? Please give some discussion on the origin of this contact resistance.

Reviewer #2 (Remarks to the Author):

This is an extremely interesting contribution highlighting the use of gated van der Pauw measurements for robust evaluation of charge transport properties of organic semiconductors. In a context where overestimation of charge carrier mobilities (through bad extractions) has started to hamper the design of the next generation of organic semiconductors, it is a pleasure to read a manuscript settling things and proposing a new reliable technique of evaluation.

Results are excellent and the experimental work was nicely and carefully done. Proper comparison of the technique with conventional extraction methods and evaluation of its limits on a material known to exhibit extremely high contact resistance have been demonstrated. Furthermore, the manuscript is very well-written and readable

Therefore I strongly recommend accepting the manuscript in Nature Communications. Following are minor comments, but they should be considered in the revision process.

1) In the abstract and conclusion, the authors claim the simple fabrication process required for the measurement. I would not put too much emphasis on forgiving alignment between the organic layer and the contacts. Of course the general averaging involved in the technique will reduce the effect of a misalignment but a decent alignment is still required as it is for conventional TFTs.

2) 7 materials are evaluated and despite giving full names and acronyms, it would be good to show their chemical structure in a figure either in the main manuscript or the SI. It will be a plus for non chemists plus some molecules are not well known like the n-type material. Moreover, except of C8-BTBT, materials' producers or suppliers have not been presented in the methods. Have you synthesized some of the molecules ? If yes, using which protocol ?

3) Figure 2c contains a lot of information and it is difficult to see properly the fitting and error bars. I would suggest to add some extra figures in the SI in order to present some isolated plots with the error bars. The use of the yellow color for the fitting is not adequate and invisible.

4) Page 7: during the explanation of the gVDP method, the use of « high » and « low » V is really confusing. "High" means positive values and "low" negatives for you but the applied absolute values are higher for the negative voltages vs positives.

5) Given the high quality of the datas and ability to properly compare several materials of the same class adequately, I am a little bit saddened by the absence of any discussion. It is obvious that a lot of efforts have been done to optimize the films and achieve large grains cf Table 3. All the elements are thus present to discuss the difference of performance that clearly demonstrates the recent works done on reducing the impact of dynamic disorder through molecular design. The manuscript is only making 3500 words at the moment and you have the room for it. It would be also important to justify the choice of the different SAMs used in the optimization process : PETS for C8-BTBT, use of Al₂O₃ + phosphonic acid SAM for the n-type material etc. Moreover, AFM pictures showing the average size of the grains for optimized films should be presented in the SI in order to be linked to the mobilities.

6) Figure 4b: despite strong contact issues, the gVDP method allows robust extraction of mobility and V_{th} for 3 different contacts. However, its is clear from the conductivity plot that sigma MoOx/Au>Au>Ag (slightly but clearly). It might be good to add a little comment on this in the main text.

7) At the end of the manuscript you present the method as very easy to realize even by scratching a continuous film in order to pattern it. You also explain that the results were in good agreement with those extracted from properly patterned devices. You should provide proofs by including in the SI the scratched devices' results and the deviation vs properly patterned devices.

8) Ref 1, 9, 11, 14, 18 and 30 do not present the complete author list

Reviewer #3 (Remarks to the Author):

The authors describe a gated van der Pauw method for measuring the mobility in organic semiconductors. Organic semiconductor parameters are notoriously difficult to measure due to low charge density and high contact resistance. The measurement method detailed in this paper overcomes the low charge density problem by using a gate and overrides the problem of contacts using the van der Pauw geometry. This method will prove useful to groups hoping to measure mobility in isotropic organic materials, and is applicable to measurement of other materials especially those that are low charge density or that are controlled using gate fields (ie ambipolar transport in graphene.)

The paper is very well written; it thoroughly covers the method and a study of contact resistance. In order to ensure that the method is employed properly, I would suggest that the authors make explicit

that the method is suited to isotropic materials, and that anisotropic materials would require a more complicated analysis (based on Montgomery method.) In addition, if there are any additional comments the authors have on using this method for materials with gate voltage dependent mobility, extremely high contact resistance, or low mobility ($< 0.1 \text{ cm}^2/\text{Vs}$) it would be appreciated.

REVIEWERS' COMMENTS:

Reviewer #1 (Remarks to the Author):

All my concerns on the previous version of the manuscript has been well addressed. Moreover, not only for my comments, but also the one raised by the other reviewer on the contact issues in the present system has been well discussed, and I feel the authors' claim very convincing. Thus I feel the manuscript ready to be published in Nature Communications.

Reviewer #2 (Remarks to the Author):

Thanks to the appropriate corrections realized by the authors, the quality of the manuscript has been significantly improved. All the additions/corrections give more weight to the robustness and reliability of this new technique. Notably the inclusion of the devices patterned using the probing needles are great assets in order to highlight the ease of implementation.

Therefore I strongly recommend accepting the manuscript in Nature Communications.

Reviewer #3 (Remarks to the Author):

Thank you to the authors for your careful consideration of the comments from reviewers. The additional information added has clarified any concerns I had, and strengthened an already strong manuscript. I recommend acceptance of the manuscript for publication in Nature Communications.

Response to referees

Dear referees, thank you for carefully reading our manuscript and for your positive feedback on our work. Here below you will find a detailed response to all of your comments.

Best regards,
The authors

Reviewer #1:

Probing the intrinsic charge carrier mobility in thin films of organic semiconductors by the gated van der Pauw method

At first, I would like to commend the authors their successful evaluation of mobility in this simple but precise analysis technique, van der Pauw method giving the real mobility of charge carriers mostly free from electrical unknown parameters originated from device structures. Especially the mobility derived from the demonstrated devices depends slightly on the contact resistances varying more than two orders of magnitude, suggesting extremely high feasibility of the present technique. Thus I am principally very positive to support publication of this manuscript in Nature Commun.

Thank you for your appreciation of our work.

Only my concerns are,

1) Definitely the value of mobility has been evaluated after careful elimination of the effects from contact resistance, etc., and on this point, I am much convinced that the value of mobility is more reliable as the “real” values than those derived from the simple TFT device configurations. However still the value is estimated eventually as the results of long-range translational motion of charge carriers among the electrodes, and even with this unique Clover-leaf like configuration of the electrodes, the range of translational motion is in the order of microns which is large enough in relation to the grain size. Thus I feel it inadequate to refer the mobility as “intrinsic”.

We agree that the use the term “intrinsic” may create confusion as the mobility extracted by the gVDP method is representative of transport over large distances in thin film that are often polycrystalline. We have therefore taken the following actions to expurge “intrinsic” from the manuscript:

- We have removed “intrinsic” from the manuscript title.
- We have defined the term μ_{ifsc} on pg. 3 as: “Therefore, a more accurate, yet simple, method is highly desirable for the proper evaluation of μ_{ifsc} the charge carrier mobility in

thin films of organic semiconductors in the high charge density accumulation layer. In this definition, μ_{tfs} characterizes the contact-independent translational motion of charge carriers across the thin film semiconductor material, over distances that may be larger than typical grain size. In this sense, μ_{tfs} encompasses extrinsic barriers to transport such as grain boundaries and therefore does not necessarily correspond to the intrinsic intra-grain charge carrier mobility of the monocrystalline semiconductor.¹⁷” where ref. 17 is new and points to a review paper on single organic crystals.

- For the mobility plagued by contacts (see Eq. 4) we have opted for the term “apparent mobility” μ_{app} rather than “effective mobility” μ_{eff} used in the previous version of the manuscript. We chose this as the term μ_{eff} is ambiguous and does not clearly point to contacts as a source of problem. Also, in the literature, effective mobility is often opposed to intrinsic mobility. So we wanted to remove reference to both μ_{int} and μ_{eff} to avoid any source of confusion.

2) Please give some parameter on the solid state structure of materials deposited on the substrate such as averaged grain sizes, crystallite orientations, etc. by microscopic observation and/or XRD, otherwise the readers cannot compare the mobility values to the others.

In order to provide benchmarks for future growth studies, we have added in Fig. S5 in the Support Information AFM and XRD characterizations of thin films of each of the seven semiconductors analyzed in this manuscript. In the main text, section 3.4 on pg. 12, we have added the following generic description of thin film morphology:

“Thin (~30 nm) films of these materials were deposited in vacuum using growth conditions optimized for electrical performance. Characterization by X-ray Diffraction and Atomic Force Microscopy is given for all films in Fig. S5 in Supporting Information. This shows that all thin films are polycrystalline with growth patterns that are beneficial to lateral charge transport: The films are composed of a mosaic of two-dimensional grains with diameters ranging from 0.5 to >5 μm , that is much smaller than lateral gVDP device dimensions. These grains present a layer-by-layer microstructure with molecules standing on their long axis, which maximizes electronic coupling between adjacent molecular cores within the plane of the layer. In most cases, this continuous two-dimensional film is covered by a dense matrix of tall elongated needles. This three-dimensional growth is symptomatic of a Stranski-Krastanov roughening transition happening during the growth.”

3) In case of Ag contact, extremely high contact resistance was observed as 40 M Ω m, but why? Please give some discussion on the origin of this contact resistance.

C8-BTBT is a very difficult semiconductor to contact. This is due to its very deep HOMO level in conjunction with a poor vertical transport of charges that increases access resistance to the channel. The mismatch between Ag work function and C8-BTBT HOMO is huge and results in the very high R_{cW} of 3 M Ω cm at $V_{\text{G}} = -60\text{V}$ (See Fig. 4a). Such high values have been previously reported for the same semiconductor (see ref. 32). We simply divide this value by the length of the gVDP injecting contact edge to estimate the resistance at injection 40M Ω in the gVDP device.

To make this point clear, we have added the following text and new references on pg. 11:

“Indeed, C₈-BTBT has a deep HOMO level of -5.8 eV that complicates band alignment at the metal/semiconductor interface and poor vertical transport properties that complicate access to the channel in the staggered device topology.^{30,31} Intercalating a thin layer of MoO_x has been previously shown to improve injection.³²”

Reviewer #2 :

This is an extremely interesting contribution highlighting the use of gated van der Pauw measurements for robust evaluation of charge transport properties of organic semiconductors. In a context where overestimation of charge carrier mobilities (through bad extractions) has started to hamper the design of the next generation of organic semiconductors, it is a pleasure to read a manuscript settling things and proposing a new reliable technique of evaluation.

Results are excellent and the experimental work was nicely and carefully done. Proper comparison of the technique with conventional extraction methods and evaluation of its limits on a material known to exhibit extremely high contact resistance have been demonstrated. Furthermore, the manuscript is very well-written and readable

Thank you for these very positive comments. Our work was indeed motivated by the realization that when the mobility in organic semiconductors climbs up and contact quality stagnates, better electrical characterization methods are absolutely required.

Therefore I strongly recommend accepting the manuscript in Nature Communications. Following are minor comments, but they should be considered in the revision process.

1) In the abstract and conclusion, the authors claim the simple fabrication process required for the measurement. I would not put too much emphasis on forgiving alignment between the organic layer and the contacts. Of course the general averaging involved in the technique will reduce the effect of a misalignment but a decent alignment is still required as it is for conventional TFTs.

You are correct that the alignment constraints are similar to those needed in the fabrication of TFTs. On the other hand they are much more relaxed than the constraints required by the fabrication of the Hall bar structure needed in the gated four point probe (gFPP) measurement. As gFPP is a direct competitor of the gVDP method, it is that fact that we wanted to stress here. But this is already addressed in section 3.5.

Both in the introduction and in the conclusion, we made this statement more general by writing: “*I- Device structure and fabrication constraints are the same as for thin film transistors, allowing easy device integration and comparison.*”

2) 7 materials are evaluated and despite giving full names and acronyms, it would be good to show their chemical structure in a figure either in the main manuscript or the SI. It will be a plus for non chemists plus some molecules are not well known like the n-type material. Moreover, except of C8-BTBT, materials' producers or suppliers have not been presented in the methods. Have you synthesized some of the molecules ? If yes, using which protocol ?

We have included the molecular structures of all organic semiconductors in Fig. S5 and we have named the source of the materials in table 3 related to the experimental section.

3) Figure 2c contains a lot of information and it is difficult to see properly the fitting and error bars. I would suggest to add some extra figures in the SI in order to present some isolated plots with the error bars. The use of the yellow color for the fitting is not adequate and invisible.

Figure S1 in the SI has been augmented with five graphs showing the σ_s vs. $V_G - V_C$ characteristic for each current I_j . All this data is overlain in Fig. 2c.

4) Page 7: during the explanation of the gVDP method, the use of « high » and « low » V is really confusing. "High" means positive values and "low" negatives for you but the applied absolute values are higher for the negative voltages vs positives.

This has been clarified.

5) Given the high quality of the datas and ability to properly compare several materials of the same class adequately, I am a little bit saddened by the absence of any discussion. It is obvious that a lot of efforts have been done to optimize the films and achieve large grains cf Table 3. All the elements are thus present to discuss the difference of performance that clearly demonstrates the recent works done on reducing the impact of dynamic disorder through molecular design. The manuscript is only making 3500 words at the moment and you have the room for it. It would be also important to justify the choice of the different SAMs used in the optimization process : PETS for C8-BTBT, use of Al₂O₃ + phosphonic acid SAM for the n-type material etc. Moreover, AFM pictures showing the average size of the grains for optimized films should be presented in the SI in order to be linked to the mobilities.

As you may have seen in our answer to Reviewer 1, we have added in the SI a broad section with systematic characterization by AFM and XRD of all thin films analyzed in this work. We also have added a generic description of film morphology in section 3.4 on pg. 12.

The goal of the present paper is to present a measurement method rather than to perform a material study. In consequence, the sole purpose of the thin films of organic semiconductors analyzed here is to demonstrate the measurement technique. We selected these materials as they are representative of the latest trends in the field of high-mobility organic small molecular semiconductors and we have directly reused optimized growth recipes that have delivered the best TFT characteristics in other works of our group. So we didn't need to perform optimization work for the present study, neither for the growth conditions nor for the self-assembled monolayers.

Clearly it would be very interesting to perform a material study using the gVDP device to probe the electrical properties of films fabricated in different conditions. Such studies, however, can quickly become complicated considering the complexity of growth processes and the difficulty of characterizing the 2D monolayers buried underneath a dense forest of 3D structures that arise in most of these materials. In consequence, we really wanted to keep a single message in this paper: The gVDP method functions well and is applicable to a broad range of materials, regardless of their performance. It would definitely be a great device to analyze systematic growth studies, but this is left to further work. On pg. 13 we have added the following comment in this sense:

“In consequence, the gVDP method is an excellent probe to systematically relate electrical performance with the morphology and microstructure of thin films of organic semiconductors. Such systematic growth studies are left to further work.”

6) Figure 4b: despite strong contact issues, the gVDP method allows robust extraction of mobility and V_{th} for 3 different contacts. However, its is clear from the conductivity plot

that $\sigma_{\text{MoOx/Au}} > \sigma_{\text{Ag}}$ (slightly but clearly). It might be good to add a little comment on this in the main text.

This slight difference in conductivities is caused by a slight V_T spread, whereas mobility stays the same between the different contacts. In order to make this clear, we have adapted the text in section 3.3. First we give a short explanation of why contact quality affects so much the V_T obtained from benchmark TFT measurements. On Pg. 11, we wrote: “*In particular, V_T is strongly affected by contact quality, especially in short channel devices. This is caused by a voltage offset set by the non-ideal contact between the reference injecting electrode and the transistor channel that is not taken into account in the gradual channel approximation model used for V_T extraction.*”^{33,34}

Then in the description of contact effect on gVDP measurements, we adapted the text as follows:

“*The values of $\mu_{\text{tfs}}^{\text{sc}}$ reported in Table 1 have little spread and are all within error of each other. The values of V_T , also in Table 1, show a slight spread that may have the same origin as the heavy V_T spread seen above in TFTs, but with a much lower magnitude since the reference electrode is not the injecting one. The independence of $\mu_{\text{tfs}}^{\text{sc}}$ from the nature of the contact material and the low spread in V_T demonstrates that the gVDP method is quite insensitive to R_c , even when charge injection completely dominates transport as in the case of Ag contacts on C_8 -BTBT.”*

7) At the end of the manuscript you present the method as very easy to realize even by scratching a continuous film in order to pattern it. You also explain that the results were in good agreement with those extracted from properly patterned devices. You should provide proofs by including in the SI the scratched devices' results and the deviation vs properly patterned devices.

More details have been added at the end of the SI along with Fig. S6.

8) Ref 1, 9, 11, 14, 18 and 30 do not present the complete author list

This is common practice for citations in Nat. Comm.: As soon as there are more than 3 authors, the list is replaced by “et al.”.

Reviewer #3 :

The authors describe a gated van der Pauw method for measuring the mobility in organic semiconductors. Organic semiconductor parameters are notoriously difficult to measure due to low charge density and high contact resistance. The measurement method detailed in this paper overcomes the low charge density problem by using a gate and overrides the problem of contacts using the van der Pauw geometry. This method will prove useful to groups hoping to measure mobility in isotropic organic materials, and is applicable to measurement of other materials especially those that are low charge density or that are controlled using gate fields (ie ambipolar transport in graphene.)

The paper is very well written; it thoroughly covers the method and a study of contact resistance. In order to ensure that the method is employed properly, I would suggest that the authors make explicit that the method is suited to isotropic materials, and that anisotropic materials would require a more complicated analysis (based on Montgomery

method.) In addition, if there are any additional comments the authors have on using this method for materials with gate voltage dependent mobility, extremely high contact resistance, or low mobility ($< 0.1 \text{ cm}^2/\text{Vs}$) it would be appreciated.

Thank you for your appreciation of our work.

At the end of section 3.4, on page 13, we have added a comment on the case of thin films with anisotropic transport properties:

“All films analyzed in this study were polycrystalline with random grain orientation and grain size much smaller than device dimensions. In consequence, all films displayed isotropic transport properties as evidenced by equal resistances R_{12} and R_{23} measured along two perpendicular edges of gVDP devices with four-fold symmetry. The gVDP method can, however, be extended to the treatment of anisotropic thin film of semiconductors such as thin organic single crystals by adapting methods previously developed for the interpretation of VDP measurements carried out on anisotropic films.^{35,36”}

And in the text immediately following, on pg. 14, we have added a comment to broaden the application range of the method, including low mobility materials and semiconductors with field dependent mobility:

“Besides the small molecular organic semiconductors studied here, the gVDP method is generally suited to characterize a wide range of materials, such as semiconducting polymers, metal oxides, 2D materials, etc. The characterization of very low mobility semiconductors is ultimately limited by the resolution of the measurement setup. Strategies to enhance current such as device downscaling and the use of stronger dielectrics can help these measurements. In the case of semiconductor/dielectric systems that display significant field-dependence of the mobility ($\gamma > 0$), a sublinear behavior of the σ_s vs. $(V_G - V_D)$ characteristic is expected. Data analysis would require the derivation of a model equivalent to Eq. 3 that still contains the mobility enhancement factor γ .”

Finally, it is difficult to predict the effect of extreme contact resistance. The case of C_8 -BTBT with Ag contact was already pretty bad and, much to our surprise, worked like a charm. Pushing R_c further, we would assume that ultimately, in order to maintain a constant I_I , the voltage V_I on the injecting electrode I would become so high that it would lead to heavy bias stress or to dielectric breakdown.